# Schema-Backed Visual Queries over Europeana and other Linked Data Resources

Kārlis Čerāns[1], Jūlija Ovčiņņikova, Uldis Bojārs, Mikus Grasmanis,
Lelde Lāce, Aiga Romāne

Institute of Mathematics and Computer Science, University of Latvia
[1]karlis.cerans@lumii.lv

**Abstract.** We describe and demonstrate the process of extracting a data-driven schema of the Europeana cultural heritage Linked data resource (with actual data classes, properties and their connections, and cardinalities) and application of the extracted schema to create a visual query environment over Europeana. The extracted schema information allows generating SHACL data shapes describing the actual data endpoint structure. The schema extraction process can be applied also to other data endpoints with a moderate data schema size and a potentially large data triple count, as e.g., British National Bibliography Linked data resource.

**Keywords:** RDF, SPARQL, Linked data, Europeana, Visual Queries, SHACL

## 1    Introduction

Europeana[1] is a cultural heritage information aggregator that provides access to millions of digitized books, films, paintings, and other types of cultural heritage objects [1] (cf. also [2]). This information is gathered from hundreds of cultural institutions and is presented in a unified user interface.

Availability of the Europeana data as a SPARQL endpoint[2] allows the interested parties to issue multiple-entity and aggregated queries over its data, as well as to integrate Europeana within the Linked data landscape, where the queries can be asked about multiple data sets simultaneously (e.g., as federated SPARQL queries).

To ask the data queries efficiently, knowing the data schema is essential, and a support of a tool that can help an end-user by suggesting a schema-based information for query completion would be a major advantage. The tools relying on some sort of a schema information for a data endpoint involve visual query systems as Optique VQs [3] and ViziQuer [4], as well as facet-based [5] and natural-language based [6] systems.

The definition of the Europeana Data model[3] (including the Mapping Guidelines) provides the description of the classes and properties that characterize the available data; however, these may not be fully in sync with the actual data structure necessary

---

[1] https://www.europeana.eu/

[2] http://sparql.europeana.eu/

[3] Europeana Data Model, https://pro.europeana.eu/page/edm-documentation

for the query creation support. A similar situation is typical for Linked data resources in general, as their provision is not necessarily schema based.

Although SPARQL has instructions for the schema retrieval from the data, running these on a large-scale endpoint such as Europeana (holding more than 2.8 billion data triples), is not always technically feasible without explicit arrangements for performance.

The novel contributions of the paper are the following:
- a generic method for extracting a moderately sized data schema, suitable for a visual query environment creation, from a large SPARQL endpoint such as Europeana or British National Bibliography (BNB)[4], and
- automated generation of SHACL [7] descriptions of the actual endpoint data structure from the extracted data schemas.

We demonstrate the visual query environment creation in the ViziQuer tool[5] [4], however, we expect that the schema information can be easily restructured also to support other use cases, as demonstrated by the SHACL description generation.

In what follows, Section 2 describes the data schema extraction process, Section 3 sketches an obtained visual query environment, Section 4 comments on SHACL generation, provides some further discussion and concludes the paper. The resources supporting the paper are available at https://viziquer.lumii.lv/examples/europeana2021.

## 2 Data Schema Extraction

A data schema shall contain the information about the classes and properties in the data model, and their interconnection. The cardinality information and the data and object property separation shall also be recorded in the schema, where possible.

Given a SPARQL endpoint, one can write queries for collecting the schema information, however, for endpoints of substantial data size (as the Europeana endpoint is), the execution of such queries may not be possible technically. To overcome the limitations (where possible), we propose the following schema extraction process:
1) determine the **list of classes**, together with the **instance count**, if possible (instance count can be asked in the joint query, or for each class separately),
2) determine the **list of properties**, together with the **triple count**, if possible (triple count can be asked in the joint query, or for each property separately),
3) determine the **subclass relation**, where possible (to check if A is a subclass of B, ask, if A and B have a joint instance, then, if none of A instances is outside B). As an example, a SPARQL query template used here would be:
   *SELECT ?x WHERE { ?x a <classA>. OPTIONAL { ?x a ?value.*
   *FILTER (?value = <classB>) } FILTER (!BOUND(?value)) } LIMIT 1*
4) count (look for existence of) **data and object triples of a property**, obtaining a (non-strict) separation of properties as (primarily) data and object properties.
5) determine the **class and property map**: for each property and each class check, if there is a class instance that is a subject of a property triple and if there is an

---

[4] https://bnb.data.bl.uk/sparql
[5] https://github.com/LUMII-Syslab/viziquer, http://viziquer.lumii.lv/

instance that is an object of a property triple; if possible, count the respective triples. If a single query over all properties fails, go class by class, if that fails, go property by property, then go by individual (class, property) pairs, if needed,

6) for a property that has data triples, compute the **property and datatype map** – find the datatypes that are types of the property triple object values,

7) compute the **minimum and maximum cardinality** (check, if any of these is 1) for each property in the context of each its subject class, as well as for the **inverse** of each property in the context of each its object class,

*8)* for each object property $p$ compute its **most important target class set** to cover as much of $p$ triple objects, as possible by a limited size class set. We order all $p$ target classes $c$ descending by their *(p,c)*-count (the count of $p$ objects that are instances of $c$), then by the $c$ size ascending. We walk through the ordered class list and mark a class important, if it contains a $p$ object that is not an instance of any class marked already as important. A SPARQL template for determining if to mark a class *<newc>* as an important target class for a property *<p>* in the presence of classes *<cc1>*, .. , *<ccn>* already marked as important, would be:
*SELECT ?y WHERE { ?x <p> ?y. ?y a <newc>.*
*OPTIONAL {?y a ?cc. FILTER (?cc=<cc1> || ?cc=<cc2> || .. || ?cc=<ccn>)}*
*FILTER (!BOUND(?cc))} LIMIT 1*
The process can be continued through all $p$ target classes, or a pre-defined important target class set size limit (e.g., 5 or 7) can be introduced. In each case we note, if the union of all important target classes found is a proper **range** of $p$ (i.e., it covers all $p$ objects).
The **most important source class set** for any property is computed similarly; we check, if it is a **domain** of $p$.

9) compute the **class-property-class triples** (pairs of subject and object classes for each property from the class/property map; this corresponds *to type-property paths* in terms of [8]), with triple count (*frequency* [8]), where possible,

10) if possible, compute **the most important target class set** for each property in the **context of each source class** and **the most important source class set** for each property in the **context of each target class** (following the schema of p.8).

The most important target/source class set and range/domain information is an essential part of the data schema for suggesting properties in a class context together with their "other end" classes. For instance, in Europeana the property *edm:rights* relates *ore:Aggregation* and *cc:License* classes, however, only 32K of 65M *edm:rights* triple objects are of type *cc:License* (the others do not have any type assertion), so it is important to offer in UI also the option to introduce the *edm:rights* property into the query without the object class information (since *cc:License* is not a range for *edm:rights*).

The schema information can be used for the visual query environment creation in ViziQuer tool [4] and for generation of the dataset description in SHACL [7].

An initial implementation of the schema extraction algorithm is available in the open source ViziQuer-related OBIS-SchemaExtractor service[6]. The supporting page provides the queries asked to retrieve the schema for Europeana (from the server log).

---

[6] https://github.com/LUMII-Syslab/obis-schemaextractor

## 3 Visual Query Environment

The data schemas obtained by the schema extraction algorithm, described in Section 2, can be loaded into the open source ViziQuer tool environment [4]. Figure 1 shows a visual query environment fragment over Europeana SPARQL endpoint, involving a class list (right) and two versions of a simple query: a direct visual rendering of Example 3 from [9] and its class-enriched variant. Both visual queries can be translated into SPARQL and executed over the Europeana SPARQL endpoint by the tool.

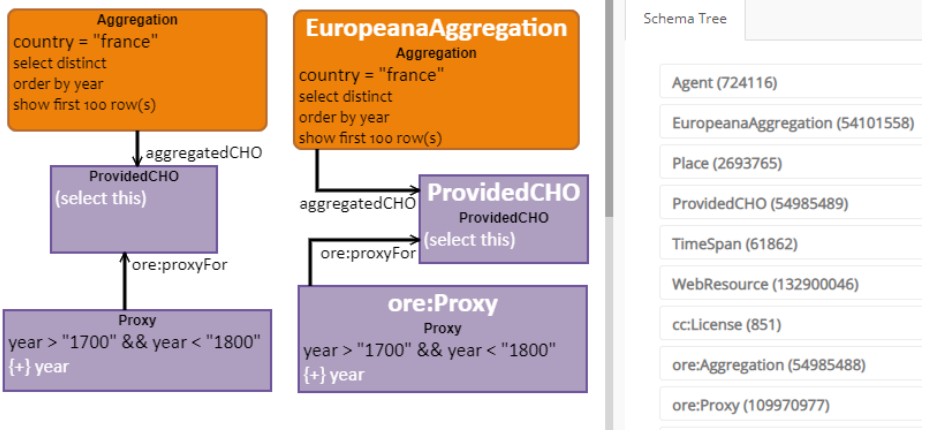

**Figure 1.** Objects provided to Europeana from the 18th century from France

The query environment provides text completion options for the class and property names in the query, including the property path construction.

The visual query presentation can be seen to have appearance benefits if compared to the textual query presentation form [9] by virtue of structuring the query across several interlinked graph nodes. The schema-based environment facilitates using of the class-enriched option that has further readability benefits, as the class names provide the conceptualization of different nodes in the query (using class-name like variable names do not always succeed in such presentation and can be even misleading).

The papers' supporting resource page provides visual presentations of SPARQL queries corresponding to [9], together with access to a live query creation environment.

## 4 Discussion and Conclusions

SPARQL query access to cultural heritage and other Linked data resources is important for enabling rich ad hoc data enquiry options. Schema information availability and visual query presentation/composition style can ease the query composition task (cf. [10]).

The generic data schema extraction process described here would allow retrieving the actual data schema that can be used to support the visual query creation over the data heavy and moderate schema size Linked data resources such as Europeana or British National Bibliography SPARQL endpoint. The actual data schemas extracted from

the endpoints can be saved also in SHACL [7] format enabling their use in other contexts, as well. There can be various level of detail the SHACL descriptions of a data set could provide. Regarding the Europeana endpoint the most informative is a manually created specification in [11]. Our contribution here is the ability to obtain the SHACL description of the data set structure (involving cardinalities) automatically from the data set. We also introduce descriptions of inverse properties that are relevant in the context of a class. The generated SHACL schemas can also be extended e.g., by the triple count information for a property in a class context, that can be useful in UI generation.

The related approaches of data schema retrieval by SPARQL queries, as e.g. [8],[12] do not reach the capability of retrieving the full Europeana data schema.

A limitation of the proposed data schema extraction process is that it would not be suitable for endpoints with very large actual data schemas, as DBPedia and Wikidata; data schema extraction and visual query support for these endpoints is further work.

## Acknowledgements

This work has been partially supported by a Latvian Science Council Grant lzp-2020/2-0188 "Visual Ontology-Based Queries".

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
