# OpenReview forum: "Schema-Backed Visual Queries over Europeana and other Linked Data Resources"
_eswc-conferences.org/ESWC/2021/Conference/Poster_and_Demo_Track — ESWC2021 P&D_

### Official Review · ~Ben_De_Meester1 · 2021-04-13
**Dual focus on contribution lowers impact of proposed data schema extraction method**

**Rating:** 6
**Confidence:** 3

**Review:**

This paper presents a data schema extraction method, and applies it to viziquer to showcase its results as a visual query environment.
For me, the contribution is a bit unclear. Quite some space is spent on the visual demo, however, I don't see that as a contribution. I would have preferred to have a more detailed description of your data schema extraction method -- as I think that is the contribution -- and what it entails, eg
inclusion of examples, the kind of conclusions you were able to make (e.g. did this automatic method showcase discrepancies with the Europeana EDM? With the SHACL shapes of Hugo Manguinhas?), etc.
In general, I'm not sure what the reader could take from this, nor whether the presented method has any soundness since it's hard to grasp based on the abstract descriptions of p2,
without clarifications on what these steps return and how these results are relevant. For example: how does the class and property map relate to the actual data schema? Is it a direct mapping? Why is this a good idea?
In general, I have the feeling the additional focus on using viziquer is either not wel argumented, or not necessary. Space that could have been put to better use by better argumenting the proposed method.

**Anonymity:**

No, I would like my review to be deanonymized.

---

### Official Review · ~Marilena_Daquino1 · 2021-04-14
**unclear innovative aspects or benefits**

**Rating:** 3
**Confidence:** 5

**Review:**

quality
- the poster focuses on the schema extraction from Europeana SPARQL endpoint and the presentation of a tool for visual query. The state of the art of data schema extraction and WYSIWYG query tools is missing. No comparison is made in terms of benefits that the proposed solution adds. The poster fails in demonstrating the potential benefit of this work.

clarity
- needs proofreading, e.g. unclear sentence in the introduction: “these are not necessarily fully in sync with the actual data structure that is necessary for the query creation support.”

originality
- authors fail in presenting the innovative aspects in both the method for schema extraction and the visual query tool

significance
- unclear who is the target of the visual tool. If this is for experts, it seems that it does not add any feature wrt other existing tools. If it is for novices, I believe it is still far from improving the readability of the queries.

cons
- unclear target of the visual tool
- unclear benefit in using this tool
- it works on small schemas

**Anonymity:**

No, I would like my review to be deanonymized.

---

### Official Review · AnonReviewer2 · 2021-04-14
**Interesting topic, but a comparison with many similar tools would be useful**

**Rating:** 7
**Confidence:** 3

**Review:**

The authors present an approach for visual query formulation deployed over the well-known Europeana cultural heritage database. The system starts with collecting metadata about the endpoint and then provides the user with an interface to define a query using this metadata. The visually defined query is then translated into SPARQL and submitted to the endpoint.

Overall, the demo is relevant for the conference, but the novelty of the approach in comparison with the state of the art could be outlined better. There are several existing systems for visual SPARQL query formulation (such as RDF Explorer or Visual SPARQL Builder) and it would be nice if the paper still mentioned the added value of the demonstrated system. If this analysis is presented elsewhere in a full paper, would still be useful to just list the main points.


**Anonymity:**

Yes, I would like my review to remain anonymous.

---

### Official Review · AnonReviewer3 · 2021-04-16
**Interesting work, but a bit preliminary and lacks clear demo scenario**

**Rating:** 5
**Confidence:** 4

**Review:**

The paper Schema-Backed Visual Queries over Europeana and other Linked Data Resources offers an approach to extract schema information from linked data and use it for creation of visual query environments.

PROS

The problem of intuitive query interfaces over linked data is important for the community and thus methods that automate construction of such interfaces are highly appreciated. A method to support such automation is presented in the paper and implemented with the help of ViziQuer (some earlier work of the authors from 2018).

CONS

The proposed method is rather simplistic and its value is not clear, e.g., whether it helps to construct any reasonable visual query interfaces. The paper also lacks a clear demo scenario that the demo attendees will be exposed to.

SUMMARY

Although the problematics raised in the paper is important, the value of the proposed solution and the demo scenario are not clear. Thus, I have to suggest to reject the paper.

**Anonymity:**

Yes, I would like my review to remain anonymous.

---

### Official Review · Program_Chairs · 2021-04-18
**Metareview: Accept (But requires clarifications)**

**Rating:** 6
**Confidence:** 5

**Review:**

This was quite a controversial paper, with both positive and negative reviews equally weighted in score. The reviewers agree that the system addresses an important problem in terms of helping users to build queries over selected sources. Europeana is also an interesting use-case for such a tool. On the other hand, much of the focus of the paper is on ViziQuer, which is the subject of a previous paper from 2018. Novel aspects of the work remain unclear versus other query builders. One aspect that appears novel is the extraction of meta-data from sources, but details are not provided, and it is not clear how the resulting data might help to build interfaces.

Overall, this is a borderline paper. We have opted to accept as it addresses an important problem, and targets an interesting use-case. We believe that the P&D session is a good venue for discussing ongoing but promising works that address important problems, and this paper would seem to fit. We request, however, that the authors carefully take into consideration the reviewers' comments, that they add the clarifications necessary (if there is insufficient space for all details, they may be provided on a webpage or repository online), and that they include a better description of the novelty of the work in relation to existing systems.

**Anonymity:**

Yes, I would like my review to remain anonymous.

---

### Decision · Program_Chairs · 2021-04-19

Accept